# The Behavioral and ERP Responses to Self- and Other- Referenced Adjectives

**DOI:** 10.3390/brainsci10110782

**Published:** 2020-10-27

**Authors:** Alexander Savostyanov, Andrey Bocharov, Tatiana Astakhova, Sergey Tamozhnikov, Alexander Saprygin, Gennady Knyazev

**Affiliations:** 1Laboratory of Differential Psychophysiology, State-Research Institute of Physiology and Basic Medicine, 630117 Novosibirsk, Russia; bocharov@physiol.ru (A.B.); s.tam@physiol.ru (S.T.); saprigyn@mail.ru (A.S.); knyazev@physiol.ru (G.K.); 2Laboratory of Psychological Genetics, Institute of Cytology and Genetics of SB RAS, 630090 Novosibirsk, Russia; 3Laboratory of Biological Markers of Human Social Behaviour, Humanitarian Institution at the Novosibirsk State University, 630090 Novosibirsk, Russia; tastahova95@yandex.ru

**Keywords:** electroencephalogram (EEG), event-related potential (ERP), processing of self-referential information, recognition of written speech, emotionally colored words

## Abstract

The aim was to investigate behavioral reactions and event-related potential (ERP) responses in healthy participants under conditions of personalized attribution of emotional appraisal vocabulary to one-self or to other people. One hundred and fifty emotionally neutral, positive and negative words describing people’s traits were used. Subjects were asked to attribute each word to four types of people: one-self, loved, unpleasant and neutral person. The reaction time during adjectives attribution to one-self and a loved person was shorter than during adjectives attribution to neutral and unpleasant people. Self-related adjectives induced higher amplitudes of the N400 ERP peak in the medial cortical areas in comparison with adjectives related to other people. The amplitude of P300 and P600 depended on the emotional valence of assessments, but not on the personalized attribution. The interaction between the attribution effect and the effect of emotional valence of assessments was observed for the N400 peak in the left temporal area. The maximal amplitude of N400 was revealed under self-attributing of emotionally positive adjectives. Our results supported the hypothesis that the emotional valence of assessments and the processing of information about one-self or others were related to the brain processes that differ from each other in a cortical localization or time dynamics.

## 1. Introduction

The way people perceive themselves and other people is a key point of their social behavior. Fritz Heider [1] suggested the theory of attribution as a method that can be used for evaluating how people perceive themselves and others. The theory proposed that attribution is a process of evaluation of characteristics or features to one-self or another person. Thus, one of the major areas of interest for researchers is the role of self-attribution in the human mind, whether it has special mechanisms of processing self-related information or not [2]. Therefore, it is of interest if there are differences between a brain’s processing of information about one-self and information about other objects, including other people? Research has proven that information somehow related to oneself can be remembered better than information related to other persons or inanimate things, which is called “self-reference effect” [2,3]. The processing of self-related information can modulate the connection between tasks and perception, memory, and decision making, as well as between steps of information processing [4]. Previous research suggested that self-referential tasks were performed faster and had higher memory assessment [2,5,6,7,8,9]. Additionally, a “familiarity effect” was found [10,11]. This effect showed that the more familiar someone was to the participant, the better the participant remembered information about them. Thus, information about the self was remembered the best. In addition, it was shown that people’s “own” things were remembered and processed better than someone else’s [12,13,14]. Furthermore, people perceive self-referenced negative information as more negative, rather than negative information referenced in regard to other people [15].

As neuroscience research presented, the differences between self-related and non-self-related tasks were found in brain processing. Additionally, it was found out that functions of certain brain areas were strongly associated with self-related tasks [16]. Functional magnetic resonance imaging (fMRI) research studies were based on the self-related experimental models containing sentences about one-self’s traits. Interestingly, self-referential processing can be differentiated from non-self through fMRI. It has been shown that during references about non-self, activity was recorded in the inferior frontal cortex, while the medial prefrontal cortex was additionally activated under the condition of introducing self-references [17]. The processing of self-related information also induced an activation of such brain structures as the cingulate cortex, precuneus and temporal–parietal cortex [18,19,20,21,22]. The event-related potentials (ERPs) can be applied to characterize the time course of neural processes involved in different stages of self-recognition or self-assessment [23,24,25]. Usually, the comparison of amplitudes of N200, P300 and P600 components of ERPs under the recognition of one-self, highly familiar and unfamiliar faces was used in such studies. According to the results of Alzueta et al. [23], the neuronal processing of self-face differed from the recognition of a highly familiar face. Lu et al. [24] revealed that ERP amplitudes were significantly greater in self-face than in a famous face spelling paradigm at the parietal (P300 and P600 components) and fronto-central (from 700 to 800 ms) cortical areas. Munoz et al. [25] have shown that personal objects elicited lower N2 and higher P3 components compared to non-personal objects. Processing of personal objects induced more extensive connectivity to N2-P3 ERP complex between cingulate/precuneus and parahippocampal gyrus with anterior cingulate than for non-personal objects.

This study has been focused on the behavioral and ERP differences during processing of self-referential and non-self-referential information. However, emotionality had additional effect on processing of information. It has been shown that emotional stimuli induce higher arousal in the subject than neutral ones [26,27], while emotionally negative stimuli induced higher arousal than emotionally positive ones [28]. This remains true for both visual and auditory stimuli. For example, Wexler et al. [29] used a dichotic test with emotional and neutral words to show that people unconsciously processed the emotional word rather than the neutral. In addition, a decrease in brain electrical activity occurs 150–300 ms after stimulus onset as a reaction to the presentation of emotionally pleasant and emotionally unpleasant stimuli in comparison to neutral stimuli [30,31,32]. It has been shown that different types of emotional stimuli may be processed by varying neural circuits. According to fMRI studies, parietal, occipital and lower temporal lobes participate in the processing of emotional pictures [26,33,34,35]. While Fossati et al. [28] showed that negative words caused more alpha power desynchronization in insula and lower parietal lobe than positive words, Morgan et al. [19] noticed that emotionally positive words caused a higher activation level in the anterior cingulate cortex in comparison with negative words. Based on a number of publications [19,26,27,28,30,31,32,33,34,35], we hypothesized that emotional (both negative and positive) assessments should elicit stronger ERP responses than neutral assessments. However, the main issue of the study was the interaction of emotional assessment and its attribution to different people, including the participant.

It was still unknown how processing of self-related information correlated with emotions (see meta-analysis of [17,36]). There were two opposite hypotheses on this. According to the first hypothesis, there were no specific neuronal processes associated with self-referential processing, but self-related information caused stronger emotions and attention. If this hypothesis was true, then differences in brain responses to emotionally different stimuli should be detected in the same brain regions and at the same time intervals as differences in responses to stimuli attributed to one-self and others. According to another hypothesis, the recognition of the stimulus’ emotional coloring and the processing of information about oneself and others were different, though related to each of the other processes that differed in localization or temporal dynamics. For example, Fossati et al. [28] reported that the tasks containing self-issues activated the right dorsomedial prefrontal cortex, not depending on the emotional coloring of words. As for positively and negatively colored words, areas out of medial prefrontal cortex were different. In our ERP study, we compared brain responses while referring adjectives with different emotional coloring to ourselves and others. We hypothesized that the use of ERP can reveal the differences between the processing of self-referential and non-self-referential information and also the interaction of the stimuli’s emotionality with the referential effect.

Another aspect of our research concerned comparing one-self with different categories of other people. Most researchers compared differences in brain activity when participants perceived one-self, close friends/relatives and unfamiliar people. It was important to note that the question of perceiving a person that participants had a conflict with remained unclear. Social interaction may occur either in a form of a conflict/competition or in a form of cooperation. Meanwhile, we did not succeed in finding papers that compared the process of self-assessment of participants with assessment of both pleasant and unpleasant persons. We hypothesized that ERP responses in evaluating an “enemy” may differ from those which evaluate oneself, and those evaluating friends or strangers.

To sum up, if a person attributed some information to themselves, they concentrated their attention harder and remembered it better. In addition, such information induced stronger activation of brain activity. Emotional valence of words also played a role in the process of remembering things, and emotional stimuli induced higher levels of brain activity than neutral stimuli, while negative information draws more attention than positive. In this study, we investigated the ERP responses to emotional words in the context of different degrees of familiarity of the participant and the attribution object. We assumed that there can be a difference in recognition of self-referenced adjectives and non-self-referenced adjectives, which can be reflected on both the behavioral data and the ERPs indices. The more familiar the attribution object was to the participant, the higher the activation of brain regions would be.

## 2. Materials and Methods

### 2.1. Participants

Fifty-nine Novosibirsk State University students, all native Russian speakers, healthy and right-handed, were invited as participants (25 men, mean age is 22.6 years, from 17 to 25 years). All participants had no history of neurological, psychiatric, or major somatic disorders. According to the self-report, they denied use of narcotic drugs or other psychoactive substances. All participant protection guidelines and regulations were followed up in accordance with the Declaration of Helsinki. The study aim was explained to all participants and they signed the informed consent. The study and the consent form were approved by the Institute of Physiology and Basic Medicine ethics committee (https://www.physiol.ru/nauka/bio) protocol № 11 from 20th October 2018.

### 2.2. Experimental Procedure

The stimuli for the study were selected in a preparatory behavioral experiment. Thirty professional linguists (teachers and students of the Faculty of Humanities of Novosibirsk State University) were invited as participants, none of them took part in the following electroencephalogram (EEG) research. Participants were asked to rate 900 randomly selected Russian adjectives on scale from −10 to +10 (where −10 was a definitely negative adjective and +10 was a definitely positive adjective) through 0 (neutral adjectives). The ratings of all the participants were averaged. The following selection criteria were employed: positive adjectives (e.g., clever, honest, sincere) should have been rated from +7 to +10, neutral adjectives (e.g., simple, calm, silent) should have been rated from −3 to +3 and negative adjectives (e.g., stupid, rude, cynical) should have been rated from −10 to −7. Intra-class correlation (ICC) analysis using two-way mixed effects model showed high level of agreement between raters (ICC = 0.94, *p* < 0.001). In total, 150 adjectives were selected: 50 positive, 50 neutral and 50 negative (Table 1). The averaged valence ratings were −8.97 ± 0.92 for negative adjectives, 0.08 ± 1.83 for neutral adjectives, and 9.31 ± 0.71 for positive adjectives. The frequency dictionary of modern Russian language (2009), section “General vocabulary”, was used to estimate the frequency of word distribution. All selected adjectives belonged to a group of common words. The adjectives from different categories were balanced in a number of syllables. The majority of selected traits referred to personality characteristics, but some referred to physical appearance (e.g., brown-eyed, blond). Almost all positive and negative adjectives described the personality traits. Most of the neutral adjectives described the physical traits (31 adjectives) and the other 19 neutral adjectives were referred to the personality traits. We compared behavioral measures and ERPs for emotionally neutral personality trait adjectives and emotionally neutral non-personality trait (physical) adjectives. However, the special statistical comparison did not reveal any differences in the effects of personality and physical traits. For this reason, the adjectives referred to personality and physical traits were combined together.

Before the beginning of the experiment, the participants were given the first instruction. They were asked to choose one person from their surroundings for each reference object: self, loved person, neutral person and unpleasant person. As a loved person, participants should have chosen a relative or a family partner the participant had positive relationships. An unpleasant person was supposed to be a person the participant had a conflict with. Neutral person should have been somebody who was not a friend, and not an enemy. Participants did not tell the examiner who exactly they had chosen as reference objects. Then, participants were asked to rate the chosen person on a scale from −5 to +5 (including zero) where −5 was very unpleasant person and +5 was very pleasant person.

At the very beginning of the experiment participants seated with their eyes closed for 2 min without any task. For each trial, the sequence of events was as follows. The reference object was chosen randomly for each trial. The participant had to sit with their eyes closed and think about the personal qualities of the person chosen as the object. Then, the participants were given second instruction: “You will be presented a list of words, every word expresses one of the human qualities. You need to assess how much these qualities are suitable for a person you have chosen. After words are presented on the screen, you need to press the button “1” if you think this person has this quality, and the button “2” if you think that this quality does not suit him.” Then, a set of adjectives was presented in a random order, each adjective preceded by a fixation cross for 500–1000 ms. For each adjective, participants had to answer if it described the object or not. Participants responded by pressing the left (“Yes”) or right (“No”) button using the index fingers of their left and right hand and the adjective instantly disappeared. Adjectives were presented until response onset and were followed by a blank interval of 3000–7000 ms. The experimental trial was presented in a random order for each participant. The set of adjectives was the same within reference objects. Totally, each participant had to answer 600 trials (150 trial per each object of attribution). Participants were tested in a quiet room. They were seated in a comfortable chair maintaining a chin rest state. Stimuli were presented centrally, font “Arial”, 6% from screen heights in white letters on black background on a screen at a distance of 60 cm from the subject. A keypad registered yes and no responses with millisecond accuracy.

### 2.3. EEG Recording

EEGs were recorded using 130 channels (128 EEG, vertical electro oculogram (VEOG), electro cardiogram (ECG)) via Ag/AgCl electrodes. The EEG electrodes were placed on 128 head sites according to the extended International 5–10% system and referred to Cz with ground at FzA. The Quik-Cap128 NSL was used for electrode fixation. The electrode resistance was maintained below 5 kΩ. Brain Products actiCHamp (GmbH, Munich, Germany) amplifiers with a 0.1–100 Hz analog bandpass filter were used for signal amplification. The sampling rate was 1000 Hz. A FASTRAK^®^ digitizer (Polhemus, Colchester, VT, USA) was used to measure the position of each electrode and the three fiduciary points (nasion and two preauricular points).

All EEG data have been deposited on the Internet-site of Institute Cytology and Genetics SB RAS (ftp://ftp.genomics1.cytogen.ru:8203).

### 2.4. Behavioral Data Processing

The reaction time (RT) used for the behavioral data analysis was recorded from the onset of the adjective presentation. Besides, the number of choices (NC) of negative, neutral, and positive adjectives was recorded. For each object of attribution, a participant could choose from 0 to 50 adjectives of each assessment’s category. For each participant, the NC was measured separately for each object of attribution and for each category of stimuli (totally 9 measures per participant). Statistical processing of behavioral results was carried out using the IBM SPSS Statistics software package. Univariate analysis of variance (ANOVA) with 2 following factors: “attribution object” (“self”, “loved person”, “neutral person” and “unpleasant person”), and “assessment” (“positive”, “neutral”, “negative”), was used for statistical comparisons. Further pairwise comparison using paired-samples t-test was applied.

### 2.5. EEG Pre-Processing and ERP Analysis

To assess changes in signal amplitude, associated with sentence onset, event-related potentials (ERPs) were calculated in the ERPLAB toolbox (https://erpinfo.org/erplab). The trials containing non-removable muscle artefacts were rejected from the analysis. For each subject, 135–150 trials with sentence onset were used. The time intervals −1.5 to +3.0 s before and after the fixation cross onset were analyzed.

EEGs were preliminary band-pass filtered in 1–40 Hz using elliptic filters. Following the suggestion of Delorme and Makeig [37], re-reference to average reference and baseline adjustment procedures were performed during data pre-processing. The time interval from −1.5 to −0.75 s before fixation cross onset was used for baseline correction. Independent component analysis (ICA) was used for correction of eye movement and eye-blinking artefacts. Firstly, the component’s weights were computed individually for each subject and each condition. The components corresponding to eye’s artefacts were disclosed by visual inspection of component sets together with electro oculoram and electro cardiogram records. Components of artefacts were removed in the pre-processing of EEGs.

Event-related potentials (ERPs), which were calculated in the ERPLAB toolbox, were used in order to assess changes in signal amplitudes, associated with word onset. After removing artifacts, we computed ERP values using the ERPLAB toolbox separately for every EEG channel, subject and experimental condition, and a low-pass IIR Butterworth filter at 15 Hz was applied to them (https://sccn.ucsd.edu/wiki/Chapter_08:_Plotting_ERP_images). In given time ranges, local corresponding minimum/maximum peaks were located, and both amplitude and latency (between a peak and epoch zero point) were measured. Local peaks were defined as points of min/max amplitude in a given range, larger (in absolute values) than at least 5 consecutive neighboring data points in both directions. All EEG channels were used to calculate the amplitudes and latencies of each selected ERP local peak. After that, maximal positive peak amplitudes (for P300 and P600), maximal negative peak amplitude (for N200 and N400), mean peak amplitude, and peak latencies in 50–200 ms (that corresponded to location of the N200 peak during the visual peak analysis), 200–400 (i.e., peak P300), 300–500 (i.e., peak N400), and 500–700 ms (i.e., peak P600) time ranges were averaged across nine scalp regions of interest (ROI: left frontal, medial frontal, right frontal, left temporal, central, right temporal, and left, medial and right parietal–occipital scalp regions) for each subject. The results of calculating the amplitude and latency of the ERP peaks were imported from ERPLAB toolbox into datasheets. Statistical processing of ERP results was carried out using the IBM SPSS Statistics software package. After that, the values of amplitudes and latencies for each local ERP peak values were used for repeated measures ANOVA with the Greenhouse–Geisser correction to test the main effects of such factors as “attribution object” (four levels: “self”, “loved person”, “neutral person” and “unpleasant person”), “assessment” (three levels: “positive”, “neutral”, “negative”), “cortical region” (nine scalp regions of interest), age, sex, and interactions between these factors. The age was added in the statistical model as a covariate, and sex was added as a between-subjects factor. The post-hoc tests (Bonferroni) were conducted separately for the brain regions if there was a significant interaction between “region” and “attribution object” or “assessment”.

## 3. Results

### 3.1. Behavioral Results

Mean (deviation) ratings given to the objects at the beginning of the experiment were: +4.4 (0.7) for the loved person, +1.2 (0.9) for the neutral person and −2.7 (1.4) for the unpleasant person. Analysis of variance showed high significance for main effect of factor “attribution object” (F(3, 177) = 10.3, *p* < 0.001, η^2^ = 0.20). Further t-test pairwise comparison showed that all the differences were significant (*p* < 0.001).

For the number of choices, the interaction of factors “attribution object” and “assessment” was significant (F(6, 354) = 95.9, *p* < 0.001, η^2^ = 0.71). Basically, for the unpleasant person, participants more often chose negative adjectives, while for the rest of the objects, positive adjectives were chosen more often (see Figure 1). Interestingly, the interaction of factors “attribution object” and “assessment” remained significant after the removal of the “unpleasant” category (F(4, 232) = 10.6, *p* < 0.001, η^2^ = 0.21). Participants assessed the loved person the most positively, while the assessment of the self was alike the assessment of the neutral person. We proceeded the pairwise comparisons “self”–“loved person” and “self”–“neutral person” and found significant interaction of factors in the first condition (F(2, 116) = 34.7, *p* < 0.001, η^2^ = 0.46), but not in the second (F(2, 116) = 0.7, *p* = 0.506, η^2^ = 0.017) (Table 2).

The same analysis was conducted with the usage of reaction time as a dependent variable. Significant main effect of the factor “attribution object” (F(3, 177) = 7.7, *p* < 0.001, η^2^ = 0.16) showed that the reaction time during adjectives attribution to the self (mean = 1.40 ± 0.15 s) and the loved person (1.29 ± 0.21 s) was shorter than during adjectives attribution to the neutral (mean = 1.54 ± 0.20 s) and unpleasant person (1.65 ± 0.15 s) (see Figure 2). The differences in reaction time between emotional assessments and interaction between assessments and object were statistically insignificant.

### 3.2. ERP Results

N200, P300, N400, and P600 peaks of ERP have been identified by visual inspection of the time–amplitude plots in nine cortical regions averaged for all experimental conditions (Figure 3). In addition, cortical topography has been inspected separately for each peak (Figure 4) to ensure that our results were consistent with standard ERP patterns for language recognition tasks. The N200 peak had maximal negative amplitude approximately 200 ms after stimuli onset. This peak has been identified in all frontal, right temporal and occipital–parietal cortical regions. The amplitude maximum for the P300 peak has been identified in all frontal and central cortical regions approximately 300 ms after stimuli onset and in the right temporal and all occipital–parietal regions about 400 ms after stimuli onset. The N400 peak with amplitude maximum of about 400 ms has been identified in all frontal and left temporal cortical areas, whereas the P600 peak had the maximal amplitude about 600 ms in the left temporal regions. The topographic distribution of all ERP components over cortical regions was not notably different among the assessments and the attribution objects.

Statistically significant effects of attribution object or assessment were revealed neither for the time latency nor for the mean amplitude of all peaks. In addition, no significant effects were revealed for the maximal amplitude of the N200 peak.

Significant interaction between the factors “region” and “assessment” was revealed for the maximal amplitude of the P300 peak (F(16, 928) = 3.65, *p* = 0.013, η^2^ = 0.051). Separate analysis for each cortical region revealed the significant main effect of factor “assessment” in the medial frontal (F(2, 116) = 3.54; *p* = 0.036, η^2^ = 0.16) and central (F(2, 116) = 3.55, *p* = 0.033, η^2^ = 0.17) cortical areas. The statistical significance of effects of assessment to P300 amplitude controlling for age and sex was higher, than without such control. Post-hoc comparison (Bonferroni test) revealed that the maximal amplitude of the P300 peak in these areas was higher for the negative than for both positive and neutral assessments (Table 3). The main effect of factor “sex” was significant for the maximal amplitude of P300 peak (F(1, 57) = 5.06, *p* = 0.028, η^2^ = 0.082). The positive peak amplitude, averaged across all cortical areas and experimental conditions, was higher for women (1.51 ± 0.16 µV) than for men (1.27 ± 0.18 µV). This effect was observed in all cortical areas. Any interactions of factor “sex” with “attribution object” or “assessment” were not revealed for P300 peak.

The significant main effect of “attribution object” was revealed for maximal negative amplitude of N400 peak (F(3, 174) = 4.11; *p* = 0.008, η^2^ = 0.21). This amplitude, averaged among all cortical regions, was the highest for self-assessments (−1.28 ± 0.09 µV) in comparison with other persons (loved person: −1.16 ± 0.08 µV; neutral person: −1.16 ± 0.08 µV; unpleasant person: −1.10 ± 0.07 µV). Interaction between the factors “attribution object” and “region” was also significant (F(24, 1329) = 5.67, *p* = 0.031, η^2^ = 0.064). Separate analysis for each cortical region revealed the significant main effect of factor “attribution object” in the medial frontal (F(3, 174) = 4.57; *p* = 0.031, η^2^ = 0.19), central (F(3, 174) = 5.55, *p* = 0.004, η^2^ = 0.16) and medial occipital–parietal (F(3, 174) = 5.34, *p* = 0.026, η^2^ = 0.18) cortical areas. Post-hoc comparisons revealed that in all of these areas the amplitude of N400 was the highest for self-assessments and did not significantly differ between other conditions. The ERP plots in medial frontal cortical area for different “attribution object” are presented on the Figure 5.

For the negative amplitude of the N400 peak, a significant interaction of the factors “sex” on “cortical region” on “attribution object” was revealed (F(24, 1368) = 2.48, *p* = 0.038, η^2^ = 0.056). Although for men and women the amplitude of the N400 peak was higher in response to words attributed to oneself, in women the difference was more pronounced in the parieto-occipital cortical area, whereas in men it was in the frontal areas. It should be noted that the statistical assessment of the three-factor interaction with a small number of participants cannot be accepted as a sufficiently reliable result. Therefore, the differences between men and women in the cortical topography of the processes associated with the recognition of self-related information and self-esteem should be additionally verified in subsequent experiments.

The main effect of the factor “assessment” for maximal negative amplitude of N400 was also significant (F(2, 116) = 6.31, *p* = 0.003, η^2^ = 0.25). The amplitude of N400 was significantly higher (Bonferoni: *p* < 0.05) for both negative (−1.20 ± 0.07 µV) and positive (−1.19 ± 0.07 µV) than for neutral (−1.14 ± 0.07 µV) assessments. The interaction between “assessment” and “region” was insignificant (*p* > 0.7). The interaction between the factor “attribution object” and “assessment” was significant only in the left temporal area (F(6, 228) = 2.15; *p* = 0.038, η^2^ = 0.063), and insignificant among all other cortical areas. The amplitude of this ERP peak was maximum in the case of attributing emotionally positive adjectives to self.

Significant interaction between the factors “region” and “assessment” was revealed for the maximal amplitude of the P600 peak (F(16, 928) = 2.46, *p* = 0.019, η^2^ = 0.067). Separate analysis for each cortical region revealed the significant main effect of factor “assessment” in the left frontal (F(2, 116) = 2.34; *p* = 0.013, η^2^ = 0.17), medial frontal (F(2, 116) = 2.35; *p* = 0.052, η^2^ = 0.16), and left occipital–parietal (F(2, 116) = 2.45, *p* = 0.012, η^2^ = 0.16) cortical areas (Table 4). In the left and medial frontal areas, the P600 amplitude for positive and negative assessments did not differ between each other, but it was significantly lower for neutral assessments. In the left occipital–parietal cortex, the amplitude of the P600 peak was maximal for positive assessments, whereas brain responses to neutral and negative assessments did not differ. Thus, the amplitude of the P600 in the three cortical regions was higher for positive than for neutral assessments. The P600’s amplitude for negative assessments was higher than that for neutral in the left and medial frontal cortex and did not differ from neutral in the occipital–parietal cortex.

### 3.3. Correlations between Behavioural and ERP Values

Two-tailed Pearson’s correlation coefficients with the Bonferroni’s correction were calculated between the reaction time for all objects, assessments, values of maximal amplitude, time latency of N200, P300, N400 and P600 peaks of ERP separately in all cortical ROI. Module of maximal negative amplitude of N400 had highly significantly negative correlation with the reaction time in medial frontal (*r* = −0.56; *p* < 0.001), central (*r* = −0.53; *p* < 0.001), and medial occipital–parietal (*r* = −0.58; *p* < 0.001) cortical regions (i.e., ERP response was stronger for a quicker reaction). Maximal positive amplitude of P300 had significantly negative correlation with the reaction time in medial frontal (*r* = −0.31; *p* = 0.011) and central (*r* = −0.35; *p* = 0.032) cortical regions. The latency of P300 significantly positively correlated with the reaction time in the left frontal (*r* = 0.36; *p* = 0.019) and the medial frontal (*r* = 0.31; *p* = 0.039) cortical regions. The latency of P600 had a positive correlation with reaction time in the left frontal (*r* = 0.35; *p* = 0.012) and left temporal (*r* = 0.36; *p* = 0.014) regions. Separated statistical analysis did not reveal any differences in correlations between behavioral and ERP values among different objects of attribution and different assessments. The most relevant for our study, significant correlations are presented on the scatter plots (see Figure 6).

### 3.4. Results Summary

The decision of attributing (or not attributing) an adjective to a person depended on the familiarity distance between the participant and the attribution object. People generally tended to attribute positive adjectives to people they love, and negative adjectives to unpleasant people. As for the ERP data, the object of attribution had a significant impact on the maximal negative amplitude of the N400 peak. The self-related adjectives induced higher amplitudes of the N400 peak in the medial frontal, central and medial occipital–parietal cortical areas in comparison with adjective referenced to other people. No differences in ERP amplitudes were revealed between a loved, neutral and an unpleasant person. The impact of emotional valence of assessments was significant for the maximal amplitude of P300 and P600 peaks. The P300 amplitude in the medial frontal and central cortical areas was higher for negative adjectives in comparison to both neutral and positive adjectives. No differences in P300 amplitudes were revealed between neutral and positive adjectives. The P600 amplitude was higher for positive than for neutral adjectives in the left and medial frontal and left occipital–parietal cortex. The P600’s amplitude for negative adjectives was the same as for positive in frontal areas. In the left occipital–parietal area the P600’s amplitude for negative adjectives was the same as for neutral ones. Additionally, the amplitude of N400 was significantly higher for both negative and positive than for neutral adjectives; this was the most clearly pronounced in the left temporal area. Reaction time negatively correlated with the amplitude of P300 and positively correlated with the P300 and P600 latency.

For our research, it was important that the effects of attribution object and emotional valence of assessments had different relations with ERP amplitudes. The amplitude of P300 and P600 depended on the emotionality of adjectives, but not on the attribution. The amplitude of N400 depended on the attribution in the medial cortex (frontal, central and occipital–parietal) and on emotionality in the left temporal cortex. The interaction between attribution object and emotionality was significant for the N400 negative amplitude only in the left temporal area. The amplitude of this ERP peak was maximum in the case of attributing positive emotional adjectives to one-self.

## 4. Discussion

The assessment of self and others as an aspect of human social behavior is a vast area of research. Two main topics, which are an emotional valence and a familiarity degree between a participant and an attribution object, have been chosen for our study. The task of attributing adjectives varying in emotional valence to the objects with a different familiarity degree to the participant was applied to examine the differences and possible interaction between these factors. Behavioral data and ERPs were analyzed.

The differences in the number of choices (NC) of neutral adjectives between the self-, loved person, neutral persons, and unpleasant persons were not significant. For positive and negative adjectives, interpersonal differences in NC were significant (see Table 2). For an unpleasant person, participants more often chose negative adjectives, while for the rest of the objects, positive adjectives were chosen more often. Participants assessed the loved person the most positively, while the assessment of the one-self was the same as the assessment of the neutral person. A possible interpretation of the similarity in negative assessments for one-self, a neutral person and a loved one was that in the Russian communicative culture it is unacceptable to assess one-self, one’s friends or strangers in a negative way in public. A negative assessment was used only when describing a person with whom there was a real conflict. In addition, in the Russian communicative culture, it is considered ethically unacceptable to praise one-self in public, while positive reviews about friends and loved people are considered socially acceptable and desired. Our results corresponded to the social typical stereotype of a Russia person, according to which a person should respond neutrally to himself (or herself) and not allow “boasting” of his/her positive qualities. However, the reaction time during adjectives attribution to the self and the loved person was shorter than during adjectives attribution to the neutral and unpleasant person. In our interpretation, the participants were not asked to take a voluntary control over reaction time while the behavioral measure NC was under their control. Therefore, the reduced reaction time when processing information about oneself and a loved one in comparison with information about a neutral and unpleasant person mainly reflected the involuntary concentration of their attention than the behavioral indicator NC.

The attribution object effect was observed for the amplitude of the N400 peak in the medial cortical areas including medial frontal and medial parietal regions. According to neurolinguistics studies, the N400 peak was mostly related to semantic expectancy [38,39,40]. Amplitude of this peak is usually higher during recognition of information, subjectively more important information to a participant. The judgements about one-self and the others were both involved in the same type of evaluation, but the self-assessment was internal in nature, while the assessment of others was external. This key difference led to the finding that self-assessment activated the medial cortex areas, including medial prefrontal cortex (MPFC), medial frontal cortex (MFC), and precuneus. Thus, MPFC, MFC, and precuneus were involved in the self-focused processes and were significantly less involved in the non-self-focused processes. This finding was consistent with studies showing greater activation of these brain regions under the conditions of describing one-self, compared to the description of other people [17,41,42].

The amplitude of N400 negatively correlated with the reaction time for all attribution objects and assessments and was higher for one-self than for others. According to [43,44], the N400 amplitude reflected the activation of working memory during recognition of words. Non-self-references may differ in a degree of familiarity between the person and someone they are making judgements about. It is related to the fact that people remember information connected to them better in comparison to general information or information connected to other people. It has been repeatedly shown that self-related nouns were remembered better than nouns without such relation [15]. Since people think they know themselves better than anyone else, it takes them little time to make judgements about themselves, as it was shown in our study. During the behavioral data analysis, we found out that the familiarity degree between the participant and the attribution object affected the reaction time. Thus, the reaction time under the conditions of the self-assessment and the assessment of the loved person was significantly faster than the reaction time under conditions of the unpleasant or neutral person assessment. This has been previously shown by Maki and McCaul et al., where there was no significant difference between the reaction time under the conditions of self-assessment and assessment of the mother or a close friend, while the reaction time for the assessment of an unknown person was significantly higher [45]. The amplitude of N400 correlated with reaction time. This ERP component was stronger for shorter behavioral reactions. Larger N400 amplitude in the medial cortex simultaneously with a faster response for self-assessment can be interpreted as a correlate of stronger involvement of memory in self-assessment processing in comparison with the assessment of others.

It is important that during the ERP data analysis we found only the differences between self-assessments and assessments of all other objects, but no differences between a loved, neutral and an unpleasant person were revealed for any ERP value. Although behavioral differences between attribution to “loved person” and “unpleasant person” were well-noticed, no such differences were detected at the brain response level. We assumed that we would find differences in ERPs when evaluating a loved person, a neutral person, or an unpleasant person. However, such differences had not been revealed. According to literature data, people originating from individualistic (so-called “Western”) cultures had much stronger activation of the medial prefrontal cortex associated with thoughts of themselves, compared with thoughts about non-themselves, including thoughts about relatives [17]. In contrast, in people from collectivist (“Eastern”) cultures, thinking about one-self causes the same levels of prefrontal cortex activation as when thinking about relatives [46,47]. A possible hypothetical explanation of our result is that the participants were representatives of industrial urban culture. Therefore they showed a brain activation pattern typical for “individualistic” culture; when the brain reaction to information about one-self differs from the reaction to information about all others, including the loved one, and reaction to information about the loved one, the neutral and the unpleasant person did not differ from each other.

The effect of assessment was revealed for the P300 and P600 amplitude. Both the amplitudes of P300 and P600 were not sensitive to the attribution effect. The P300 amplitude in the medial frontal and central area was the strongest for negative assessments independently to an object of attribution. According to many studies, frontal P300 reflected voluntary attention to external stimuli or own acts [48,49,50]. This result could be interpreted as an index of stronger concentration of attention of negative information. In contrast to the P300, the frontal P600′s amplitude distinguished emotional (both positive and negative) assessments from neutral ones. In our study, the P600 effects of emotional assessment were located in the left and medial frontal and left temporal area including Broca’s and Wernicke’s areas. The P600 peak usually reflected the syntactic processing of words [40,51,52], but also related to emotional processing of stimuli [53,54]. It is possible to interpret this result as an index of stronger involvement of linguistic-related brain areas in the processing of emotional words.

The interaction between the attribution object effect and the effect of assessment was observed for the N400 peak in the left temporal area. The maximal amplitude of N400 was revealed under self-attributing of emotionally positive adjectives. According to some studies [55,56], the left temporal cortex was an area responsible for contextual analysis of information and establishing of relations between words and context. It is possible to assume that this cortical area participated in the synthesis of information concerning personal attribution of words and its emotional features.

More advanced analysis of behavioral and neurophysiological effects of self-referential information and stimulus emotionality is a topic for our future investigation.

## 5. Limitations

In this study, the influence of inter-individual differences in the psychological personality traits of participants on the processing of self-referential information was not observed. This topic remains a subject for future investigations.

In this study, the differences between behavioral and ERP’s effect of adjectives referred to the personality and physical traits were not revealed. There was no statistically significant difference between the effects of neutral personality trait and neutral non-personality trait adjectives. Perhaps the reason was that only all non-personality trait adjectives were emotionally neutral. Another possible reason was that emotional adjectives (all personality traits) contrasted with neutral ones, and differences within the neutral adjectives’ group were invisible to the subjects. Most likely, there are very few personal trait adjectives that did not cause emotions. All selected personality trait adjectives were emotionally colored either positively or negatively. Comparison between emotionally charged personality trait adjectives and non-personality adjectives was indeed a very interesting topic to study. However, in our experimental model, it was very difficult to pick a non-trait emotional adjective to describe a person. This is an interesting task, but it requires a completely different experiment. Perhaps it will be a topic for our new research.

The reaction time during adjective attribution to the self and the loved person was shorter than during adjective attribution to the neutral and unpleasant person. However, any differences in the ERP patterns under attribution of adjectives to the loved person, the neutral person and the unpleasant person had not been revealed. Therefore, the mechanism of this behavioral difference is still unclear.

## 6. Conclusions

Our study supported the hypothesis of the difference between brain mechanisms of processing of self-referential information and evaluation of stimulus emotionality. The self-referential effect has been found for the N400 peak in the medial cortex, including the medial frontal and the medial parietal cortex. This effect is well-aligned with fMRI results about the role of MPFC and precuneus in the processing of self-referential information. ERP responses in the frontal cortex also varied for the adjectives with different emotionality, but these differences concerned the P300 and P600 peaks. It can be noted that the ERP responses in the left frontal cortex varied for adjectives with different emotionality, but were independent of the attribution object, while the central and temporal cortex were not sensitive to emotionality but depended on attribution. Thus, ERP components associated with processing of self-referential information and emotional valence of assessments varied in both cortical localization and time dynamics. However, the left temporal N400 peak showed dependence on both factors which corresponded to viewing this area as a place associated with complex semantic analysis of information. We have not confirmed the hypothesis that the assessment of an “unpleasant person” should be different from a neutral or a loved person. As for behavioral data, reaction times of a “loved person” were similar to reaction times under self-estimates, but in ERP values, the brain responses on self-related adjectives significantly differed from responses on adjectives related to all other types of persons.

## Figures and Tables

**Figure 1 brainsci-10-00782-f001:**
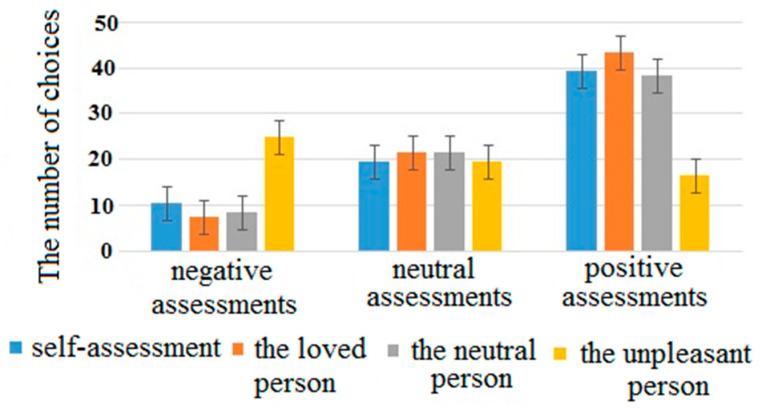
The number of choices for different attribution’s objects and assessments averaged across all participants.

**Figure 2 brainsci-10-00782-f002:**
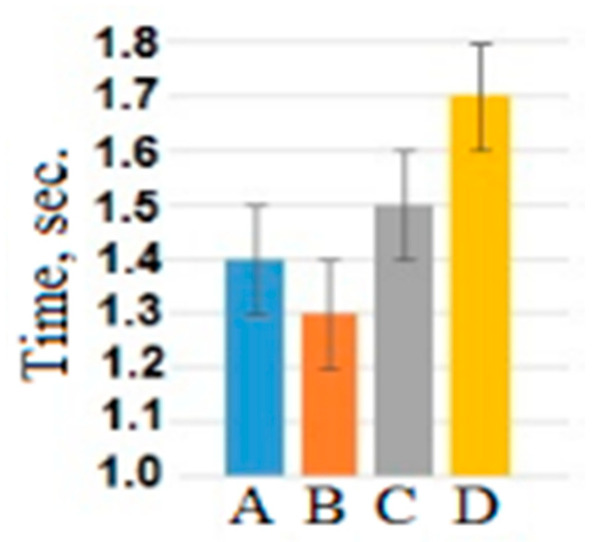
The reaction time averaged across all participants and assessments for different attribution objects: A—self-assessment, B—the loved person, C—the neutral person, D—the unpleasant person.

**Figure 3 brainsci-10-00782-f003:**
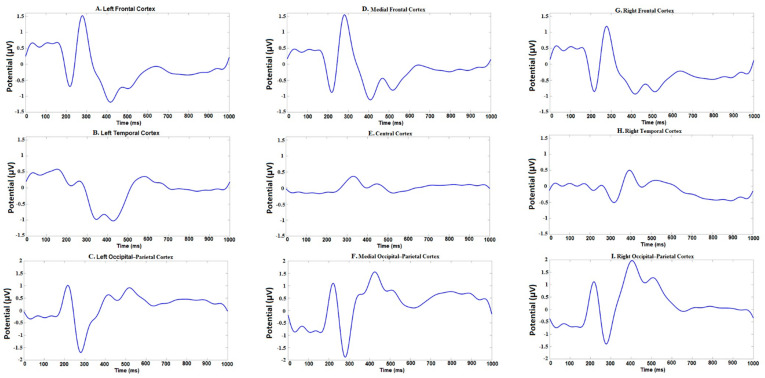
Time-amplitude plots of event-related potential (ERP) in 9 cortical regions of interest averaged across all participants.

**Figure 4 brainsci-10-00782-f004:**
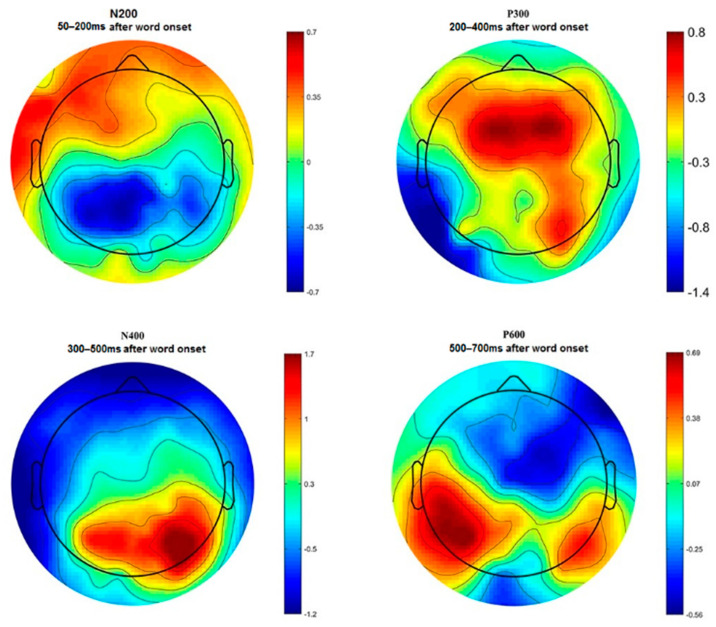
Topographical distribution of ERP amplitude averaged across all participants for N200, P300, N400, and P600 peaks.

**Figure 5 brainsci-10-00782-f005:**
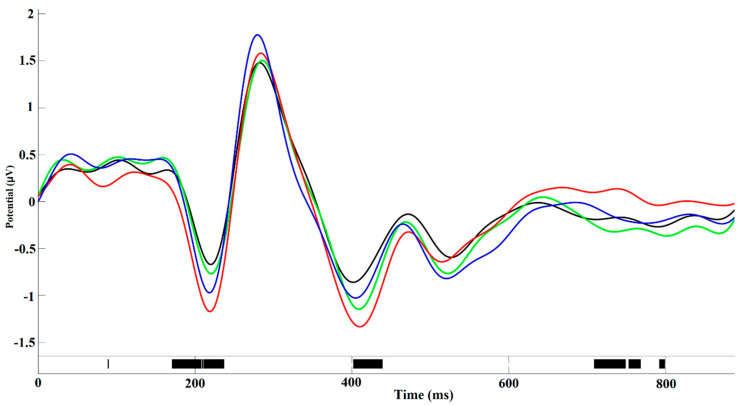
Amplitude–time ERP plot in medial frontal cortical area averaged across all assessments of one-self (red line), loved person (green line), neutral person (blue line) and unpleasant person (black line). The significant inter-object differences (permutation test) were revealed for the time intervals in about 150–250 ms, 400–450 ms, and 700–800 ms.

**Figure 6 brainsci-10-00782-f006:**
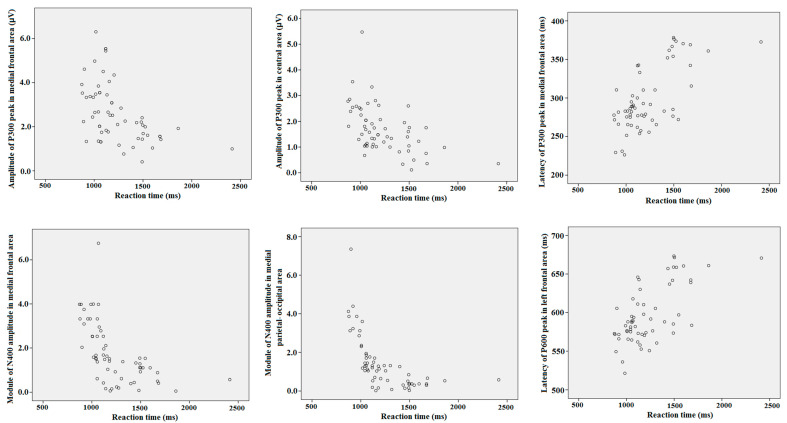
The scatter plots for correlations between reaction time and ERP peaks latency and amplitude in different cortical areas.

**Table 1 brainsci-10-00782-t001:** Verbal stimuli from the different categories with the valence ratings in a preparatory behavioral experiment.

Negative Adjectives (Russian/English)	Valence Rating	Neutral Adjectives (Russian/English)	Valence Rating	Positive Adjectives (Russian/English)	Valence Rating
тyпoй (stupid)	−8.95	лыcый (bald)	−0.53	yмный (clever)	10.00
глyпый (silly)	−8.95	poбкий (timid)	−1.97	яpкий (bright)	10.00
гpyбый (rude)	−10.00	низкий (short)	−0.53	милый (darling)	8.95
лживый (false)	−9.47	тиxий (silent)	2.63	чиcтый (clean)	8.95
жaдный (greedy)	−10.00	pыжий (red)	1.05	мyдpый (wise)	10.00
нaглый (insolent)	−10.00	плoтный (tight)	2.11	cмeлый (brave)	9.47
пoдлый (sneaky)	−10.00	гopдый (proud)	−2.63	дoбpый (kind)	10.00
xитpый (cunning)	−8.30	мeлкий (small)	−1.05	быcтpый (fast)	7.37
жaлкий (miserable)	−9.47	cтapый (old)	−2.11	вepный (faithful)	10.00
злoбный (vicious)	−8.95	xpyпкий (fragile)	1.05	чecтный (honest)	9.47
мpaчный (gloomy)	−8.85	cтpaнный (weird)	−1.05	cтoйкий (resistant)	7.37
нepвный (nervous)	−10.00	пpocтoй (simple)	1.05	aктивный (active)	9.47
льcтивый (flattering)	−7.89	бoльшoй (big)	1.05	cкpoмный (modest)	7.32
yгpюмый (moody)	−7.89	кpyпный (large)	0.00	yпopный (persistent)	10.00
циничный (cynical)	−7.32	блeдный (pale)	−2.11	щeдpый (generous)	9.47
зaнyдный (boring)	−8.95	oбычный (usual)	−0.53	дyшeвный (soulful)	8.95
cтpaшный (scary)	−7.89	pyмяный (ruddy)	1.05	кpacивый (beautiful)	8.95
ypoдливый (ugly)	−7.89	выcoкий (high)	2.63	лacкoвый (tender)	8.95
чepcтвый (callous)	−10.00	aзapтный (gambling)	−1.05	нaдeжный (reliable)	9.47
визгливый (shrill)	−8.95	кyдpявый (curly)	2.63	тoлкoвый (sensible)	8.95
дepгaный (twitchy)	−9.47	oдинoкий (lonely)	−3.00	гpaмoтный (literate)	10.00
peвнивый (jealous)	−7.47	дoтoшный (meticulous)	−2.11	иcкpeнний (sincere)	10.00
бoязливый (fearful)	−7.47	cкpытный (secretive)	−2.11	oдapeнный (gifted)	10.00
кpикливый (loud)	−9.47	пoджapый (lean)	3.00	paдocтный (glad)	9.47
пeчaльный (sad)	−7.27	шyтливый (playful)	3.00	cпocoбный (capable)	8.95
φaльшивый (fake)	−9.47	бoлтливый (chatty)	−2.63	aккypaтный (neat)	8.95
xoлoдный (emotionally insensitive)	−7.89	cпoкoйный (calm)	3.00	yлыбчивый (smiling)	9.47
oбидчивый (touchy)	−8.95	зaypядный (ordinary)	−2.63	вынocливый (tough)	8.95
плaкcивый (tearful)	−9.47	нeвзpaчный (featureless)	−3.00	paзyмный (reasonable)	9.47
нaдмeнный (haughty)	−10.00	глaзacтый (goggle-eyed)	1.05	вдyмчивый (thoughtful)	8.42
пaccивный (passive)	−7.32	мoлчaливый (silent)	0.00	гpaциoзный (graceful)	7.89
cкoвaнный (uptight)	−7.84	бeззaбoтный (carefree)	0.53	зaбoтливый (caring)	10.00
кoвapный (insidious)	−8.42	бeззacтeнчивый (impudent)	−1.58	кyльтypный (cultured)	8.95
вyльгapный (vulgar)	−8.95	бeлoкoжий (white)	3.00	кpeaтивный (creative)	9.47
вpaждeбный (hostile)	−10.00	бeлoкypый (blond)	2.63	пopядoчный (decent)	10.00
нeyклюжий (clumsy)	−7.37	близopyкий (short-sighted)	−0.53	пpeлecтный (lovely)	9.47
кaпpизный (capricious)	−10.00	бoльшeглaзый (big-eyed)	1.05	cпopтивный (athletic)	8.42
ничтoжный (worthless)	−9.47	гpoмaдный (enormous)	1.05	yникaльный (unique)	9.47
oтвpaтный (disgusting)	−10.00	пpизeмлeнный (down to earth)	−1.05	элeгaнтный (elegant)	9.47
бeздyшный (heartless)	−9.47	длиннoнocый (long-nosed)	−0.53	энepгичный (vigorous)	8.95
нeoпpятный (untidy)	−9.47	длиннopyкий (long-armed)	0.53	гapмoничный (harmonious)	10.00
бeзoбpaзный (ugly)	−9.47	дoлгoвязый (lanky)	−2.11	oбaятeльный (charming)	9.47
бeздapный (dull)	−9.47	кopeнacтый (thickset)	0.53	oбщитeльный (sociable)	8.95
бoлeзнeнный (painful)	−7.89	зyбacтый (toothy)	−0.53	кoмпeтeнтный (competent)	10.00
зaвиcтливый (envious)	−9.47	кapeглaзый (brown-eyed)	1.58	cимпaтичный (pretty)	9.47
дecпoтичный (despotic)	−8.95	cpeднepocлый (with mild growth)	1.05	тaлaнтливый (talanted)	10.00
нeaдeквaтный (inadequate)	−8.42	кapтaвый (having a burr)	−1.58	cпpaвeдливый (fair)	10.00
aгpeccивный (aggressive)	−10.00	кoнoпaтый (freckled)	0.53	cтapaтeльный (diligent)	9.47
пcиxoвaнный (nutty)	−10.00	кpyглoлицый (full-faced)	0.53	oбpaзoвaнный (educated)	9.47
oтвpaтитeльный (disgusting)	−10.00	длиннoвoлocый (long-haired)	2.63	peшитeльный (determined)	10.00

**Table 2 brainsci-10-00782-t002:** The average number of choices for different attribution’s objects and assessments.

Attribution Object	Negative Assessment	Neutral Assessment	Positive Assessment
Self	10.5 ± 4.0	19.5 ± 3.5	39.5 ± 3.0
Loved person	7.5 ± 3.5	21.5 ± 4.0	43.5 ± 3.0
Neutral person	8.5 ± 4.0	21.5 ± 3.5	38.5 ± 3.0
Unpleasant person	25.0 ± 4.0	19.5 ± 4.0	16.5 ± 3.5

**Table 3 brainsci-10-00782-t003:** The differences in the P300 maximal amplitude (in µV) between various assessments among cortical regions. Significantly different values are in bold.

Region of Interest	Negative Assessment	Neutral Assessment	Positive Assessment	*p*-Level between Negative and Neutral Assessments	*p*-Level between Negative and Positive Assessments	*p*-Level between Neutral and Positive Assessments
Left frontal	1.83 ± 0.16	1.78 ± 0.16	1.75 ± 0.15	0.58	0.58	0.58
Medial frontal	**1.94 ± 0.15**	**1.76 ± 0.15**	**1.76 ± 0.16**	**0.043**	**0.051**	0.60
Right frontal	1.54 ± 0.16	1.50 ± 0.15	1.50 ± 0.15	0.84	0.68	0.74
Left temporal	0.76 ± 0.11	0.80 ± 0.12	0.79 ± 0.11	0.59	0.38	0.56
Central	**1.45 ± 0.11**	**1.37 ± 0.11**	**1.54 ± 0.11**	**0.042**	**0.043**	0.59
Right temporal	1.09 ± 0.12	1.07 ± 0.09	1.17 ± 0.12	0.12	0.10	0.12
Left occipital–parietal	0.85 ± 0.16	0.87 ± 0.18	0.84 ± 0.17	0.88	0.89	0.76
Medial occipital–parietal	1.40 ± 0.18	1.43 ± 0.20	1.39 ± 0.19	0.80	0.79	0.65
Right occipital–parietal	1.68 ± 0.20	1.76 ± 0.19	1.81 ± 0.21	0.15	0.14	0.60

**Table 4 brainsci-10-00782-t004:** The differences in the P600 maximal amplitude (in µV) between various assessments among cortical regions. Significantly different values are in bold.

Region of Interest	Negative Assessment	Neutral Assessment	Positive Assessment	*p*-Level between Negative and Neutral Assessments	*p*-Level between Negative and Positive Assessments	*p*-Level between Neutral and Positive Assessments
**Left frontal**	**0.62 ± 0.10**	**0.50 ± 0.10**	**0.59 ± 0.11**	**0.03**	0.58	**0.04**
**Medial frontal**	**0.69 ± 0.11**	**0.55 ± 0.18**	**0.65 ± 0.12**	**0.05**	0.35	**0.05**
Right frontal	0.36 ± 0.09	0.37 ± 0.09	0.35 ± 0.10	0.35	0.48	0.35
Left temporal	1.05 ± 0.12	1.02 ± 0.12	0.99 ± 0.13	0.52	0.56	0.56
Central	0.52 ± 0.10	0.55 ± 0.09	0.57 ± 0.09	0.61	0.58	0.59
Right temporal	0.74 ± 0.10	0.74 ± 0.10	0.74 ± 0.10	0.99	0.89	0.87
**Left occipital–parietal**	**1.13 ± 0.13**	**1.18 ± 0.14**	**1.29 ± 0.13**	0.12	**0.03**	**0.02**
Medial occipital–parietal	0.92 ± 0.14	0.96 ± 0.12	1.02 ± 0.13	0.29	0.31	0.30
Right occipital–parietal	1.08 ± 0.16	1.13 ± 0.16	1.18 ± 0.16	0.32	0.32	0.30

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
