# Peer review of "The Behavioral and ERP Responses to Self- and Other- Referenced Adjectives"

_brainsci, 2020, doi:10.3390/brainsci10110782_

Round 1

Reviewer 1 Report

It is in the nature of papers such as this, with well-focused but finite goals to narrow the focus to such a degree that the paper blends into the literature.  I was therefore happy to see the authors address the issue of individualistic versus more collective social structures near the end of the paper.  While it is clearly important to stay within the boundaries of the research, I would encourage the authors to examine this area in greater detail in forthcoming research.

I do not work directly in the field the authors represent, but my own work, some of which is more historical in nature, crosses over theirs, and I found their work informing my own thinking in valuable ways.  I look forward to reading more of their material.

Author Response

Thanks for your interest in our research. In the revised version, we tried to make our paper more focused on the neurophysiological results. English has also been corrected.

Reviewer 2 Report

The article "The Behavioral and ERP Responses to Self- and 2 Other- Referenced Adjectives" aims at investigating the possible brain processing dissociation of personalized attribution depending its emotional valence. The aim of the study is interesting and timely. The study has several strengths such us the large sample size and the number of electrodes registered. However, several shortcomings dampen my enthusiasm about the study.

Main concerns:

  1. The reader drowns in the number of statistical tests and post-hoc analyses being conducted: Whereas the analyses on behavioral data is mostly straight forward, the analyses on neural responses gets a bit out of hand. In addition, more details are needed to understand the procedure followed to analyze the data. Therefore, please give more details regarding data preprocessing. For example, data were Re-referenced to what? Average reference, mastoids… ERPs were baseline corrected from-to? How was analyzed the latency of the peaks? Please put a reference for this step of the EEG analysis “cutoff filter at 15 Hz was applied to them”. Did you used all electrodes for analyses?
  2. Related to this, and more importantly, it is difficult to follow the statistical analyses. Did you use an external program such us SPPS to analyze the data or did you use ERPlab? If the second, please give some information about the analyses that perform ERPlab. With the current information, it seems that analyses were too explorative, making uncountable comparisons, in each of the nine brain regions and across all time windows, as well as, correlations between many variables. Has ERPlab anyway of controlling for this? Given this amount of analyses, Bonferroni correction is needed (if you did it, it is not stated in the text). A way to reduce the amount (and get stronger results) would be to conduct MANOVAs including the factor “laterality” (left, right and medial) within the analysis or/and reduce the number of explored cortical regions following the literature.
  3. Please only conduct post-hoc tests separately for the brain regions if there is a significant interaction between "region” and “attribute” or “emotion”.
  4. Please include a sample figure showing in one graph the ERPs for the four attribution conditions (one for pleasant, one for unpleasant and one for neutral), in the scalp region showing most differences.
  5. Please include a scatter plot of significant correlations.
  6. How were correlations performed? Which was the new p after Bonferroni correction.
  7. The authors should also include effect sizes of the results.
  8. Why was age introduced as a factor? Please specify the age range of the sample. It was a covariable? Same for sex. If you are interested in sex differences, it should be introduced as an inter-subject factor.
  9. Overinterpretation of findings about brain localization: for example, “This key difference led to find that self-assessment activated the medial cortex areas, including medial prefrontal cortex (MPFC), medial frontal cortex (MFC), and precuneus”. It is difficult to know the source of EEG data with the performed analyses. In this regard, it could be interesting to make a source analysis of the data.

Minor:

  • Please put publication following the author, for example in line 82 Fossati et al., (28)
  • Line 87: parenthesis is missing
  • Line 104: Confuse sentence, please rephrase
  • Please put an hypothesis about the emotional modulation following literature
  • Please, describe the instructions given to the participants in the validation procedure of the stimuli. Did they know that these adjectives would be referred to a person? For example “simple” can be neutral in a neutral context but negative when attributed to a person (not clever). Maybe in Russian it as not this connotation. Anyway, the instruction is important.
  • Line 191: There were two or three factors? You say three but describes just two…

Author Response

We are grateful for your very helpful and detailed remarks. We revised of the manuscript according to your suggestions.

1) data were Re-referenced to what?

To average reference. See line 213.

ERPs were baseline corrected from-to?

Time interval from −1.5 to −0.75 s before fixation cross onset was used for baseline-correction. See lines 214-215.

How was analyzed the latency of the peaks?

In given time ranges local corresponding minimum/maximum peaks were located, and both amplitude and latency (between a peak and epoch zero point) were measured. Local peaks were defined as points of min/max amplitude in a given range, larger (in absolute values) than at least 5 consecutive neighboring data points in both directions. All EEG channels were used to calculate the amplitudes and latencies of each selected ERP local peak. See lines: 224-228.

Please put a reference for this step of the EEG analysis “cutoff filter at 15 Hz was applied to them”.

Thank you for this comment. See lines 223-224: a low-pass IIR Butterworth filter at 15 Hz was applied to them (https://sccn.ucsd.edu/wiki/Chapter_08:_Plotting_ERP_images). 

Did you used all electrodes for analyses? Yes, see line 228-229.

2) Related to this, and more importantly, it is difficult to follow the statistical analyses. Did you use an external program such us SPPS to analyze the data or did you use ERPlab? If the second, please give some information about the analyses that perform ERPlab. With the current information, it seems that analyses were too explorative, making uncountable comparisons, in each of the nine brain regions and across all time windows, as well as, correlations between many variables. Has ERPlab anyway of controlling for this? Given this amount of analyses, Bonferroni correction is needed (if you did it, it is not stated in the text). A way to reduce the amount (and get stronger results) would be to conduct MANOVAs including the factor “laterality” (left, right and medial) within the analysis or/and reduce the number of explored cortical regions following the literature.

Many thanks for your recommendations. See line 235-243.

The results of calculating the amplitude and latency of the ERP peaks were imported from ERPlab toolbox into datasheets. Statistical processing of ERP results was carried out using the IBM SPSS Statistics software package. After that, the values of amplitudes and latences for each local ERP peak values were used for repeated measures ANOVA with the Greenhouse-Geisser correction to test the main effects of such factors as “attribution object” (four levels: “self”, “loved person”, “neutral person” and “unpleasant person”), “assessment” (three levels: “positive”, “neutral”, “negative”), “cortical region” (nine scalp regions of interest), age, sex, and interactions between these factors. The age was added in the statistical model as a covariate. The post-hoc tests (Bonferroni) were conducted separately for the brain regions if there was a significant interaction between "region” and “attribution object” or “assessment”.

3) Please only conduct post-hoc tests separately for the brain regions if there is a significant interaction between "region” and “attribute” or “emotion”.

It was done.

4) Please include a sample figure showing in one graph the ERPs for the four attribution conditions (one for pleasant, one for unpleasant and one for neutral), in the scalp region showing most differences.

It was done. See Figure 5.

5) Please include a scatter plot of significant correlations.

It was done. See Figure 6.

6) How were correlations performed? Which was the new p after Bonferroni correction.

It a previous version of manuscript the correlations presented after Bonferroni correction. See line 374.

7) The authors should also include effect sizes of the results.

It was done.

8) Why was age introduced as a factor? Please specify the age range of the sample. It was a covariable? Same for sex. If you are interested in sex differences, it should be introduced as an inter-subject factor.

The age range was specify. See lines 123 and 242. In fact, we didn't find any effects of age. Our participants differed little in age. We included this factor in the analysis as another reviewer insisted on it. We found some interesting effects of sex. However, we did not discuss them in this article, since it turns out to be too difficult for the reader.

9) Overinterpretation of findings about brain localization: for example, “This key difference led to find that self-assessment activated the medial cortex areas, including medial prefrontal cortex (MPFC), medial frontal cortex (MFC), and precuneus”. It is difficult to know the source of EEG data with the performed analyses. In this regard, it could be interesting to make a source analysis of the data.

It is really an extremely interesting topic for research. Unfortunately, we are limited by a size of article. Now we are preparing for publication new results of a joint EEG-fMRI study in the same experimental paradigm. In this study, we specifically focused on finding sources of brain activity under recognition of self-related information.

Minor:

Please put publication following the author, for example in line 82 Fossati et al., (28)

Thanks. It was done.

Line 87: parenthesis is missing

corrected

Line 104: Confuse sentence, please rephrase

corrected. See lines 106-108.

Please put an hypothesis about the emotional modulation following literature

See lines: 83-86.

Please, describe the instructions given to the participants in the validation procedure of the stimuli. Did they know that these adjectives would be referred to a person? For example “simple” can be neutral in a neutral context but negative when attributed to a person (not clever). Maybe in Russian it as not this connotation. Anyway, the instruction is important.

Thanks. See lines 160-162 and 170-173.

Line 191: There were two or three factors? You say three but describes just two…

corrected.Of course just two.

Round 2

Reviewer 2 Report

Thank you for your  changes. I think the paper has improved and it is almost ready. Here the last points:

Please add a legend in figure 5.

Please clarify if you used sex as an inter-subject factor or as a covariate in the methods (you say that age was a covariate, but nothing about sex). If you have significant effects regarding sex you should mention them (at least in supplementary material) or say that they will be included in another publication (if this is the case) or do not include the factor... If you do not mention it seems that sex is not significant at all, which is not true. In any case, I would like to see these results. ¿Do they modify your conclusions?

Author Response

Thanks for your help in a revison of our manuscript. Your suggestions are really valuable to us.

1) Please add a legend in figure 5.

It was done. See lines 338-341.

2) Please clarify if you used sex as an inter-subject factor or as a covariate in the methods (you say that age was a covariate, but nothing about sex).

See line 242.

3) If you have significant effects regarding sex you should mention them (at least in supplementary material) or say that they will be included in another publication (if this is the case) or do not include the factor... If you do not mention it seems that sex is not significant at all, which is not true. In any case, I would like to see these results.

See lines 308-312 and 327-335.

4) ¿Do they modify your conclusions?

No.